# The role of age and gender in the relationship between personality traits, quality of life, and decision-making about orthognathic surgery— A cross-sectional study

Renata Vidakovic[1], Martina Zigante[2], Jeta Kelmendi[3]*, Stjepan Spalj[2,4]

**1** Private Orthodontic Practice, Zagreb, Croatia, **2** Faculty of Dental Medicine, Department of Orthodontics, University of Rijeka, Rijeka, Croatia, **3** Alma Mater Europea Campus College Rezonanca, Prishtina, Kosova, **4** Faculty of Dental Medicine and Health, Department of Dental Medicine, J. J. Strossmayer University of Osijek, Osijek, Croatia

* jeta.kelmendi@rezonanca-rks.com

## Abstract

### Objective

To explore how age and gender shape the relationship between personality traits, quality of life (QoL), and a patient's decision to undergo surgery for correction of dentofacial deformity.

### Methods

In a cross-sectional study, 108 individuals aged 14–53 years (median age 18 years; interquartile range 17–26) with a moderate to very great need for surgery according to the Index of Orthognathic Functional Treatment Need were assessed. There were 43% adolescents (≤17 years) and 68% females. Participants completed validated questionnaires measuring personality characteristics (Big Five Inventory, Multidimensional Perfectionism Scale, and Self-Esteem Scale), along with the Orthognathic Quality of Life Questionnaire. The decision to accept or refuse surgery was recorded.

### Results

Overall, 51% of patients accepted surgery (48% of females and 57% of males). Adults were more likely to accept surgery than adolescents (61% vs. 37%; p = 0.019, V = 0.241; odds ratio 2.7; 95% confidence interval 1.2–5.9). In adults, lower self-esteem (SE) (p = 0.034, r = 0.270) and higher perfectionism (p = 0.012; r = 0.320), particularly concern over mistakes (CM) (p < 0.001; r = 0.469) and personal standards (p = 0.004; r = 0.370), were associated with acceptance. Among adolescents, personality traits showed no significant effect. However, in both age groups, impaired oral function (OF) (p = 0.013; r = 0.368 for adolescents and p = 0.006; r = 0.348 for

**Data availability statement:** All relevant data are within the paper and its Supporting information files. The anonymized minimal dataset underlying the results presented in this study is publicly available in the institutional repository of the University of Rijeka at the following link: https://urn.nsk.hr/urn:nbn:hr:271:110097.

**Funding:** The author(s) received no specific funding for this work.

**Competing interests:** The authors have declared that no competing interests exist.

adults) and greater facial esthetics concern (FE) ($p = 0.018$; $r = 0.350$ and $p = 0.015$; $r = 0.310$) influenced surgery acceptance. Males who accepted surgery had higher awareness of deformity than those who refused ($p = 0.006$; $r = 0.463$), while females who accepted surgery had lower SE ($p = 0.044$; $r = 0.235$). In both males and females FE was higher in those who accepted surgery ($p = 0.009$; $r = 0.439$ for males and $p = 0.006$; $r = 0.320$ for females), OF ($p < 0.001$; $r = 0.597$ and $p = 0.003$; $r = 0.342$), and CM ($p = 0.030$; $r = 0.367$ and $p = 0.006$; $r = 0.325$).

## Conclusions

The relationship between personality traits, QoL, and the decision to undergo orthognathic surgery was more strongly influenced by age than by gender. OF and FE influenced acceptance across all age and gender groups, while personality-related factors were particularly relevant among adults.

## Introduction

Dentofacial skeletal deformities, defined by pronounced discrepancies in intermaxillary and occlusal relationships, affect approximately 20% of the population and typically require a comprehensive, interdisciplinary treatment approach [1]. These imbalances can impair fundamental life functions such as speech, mastication, and breathing. Even when not functionally limiting, such deformities often lead to reduced disease-specific quality of life (QoL) due to aesthetic and psychosocial concerns [2].

QoL is a multidimensional construct encompassing physical, emotional, social, and material well-being, along with aspects of personal development and daily functioning. The concept of oral health-related quality of life gained prominence in the 1980s when the broader impacts of oral health conditions, particularly their psychological and socioeconomic consequences were increasingly recognized [3].

Although individuals with dentofacial deformities may not report pain or perceive themselves as ill, they frequently express dissatisfaction with their facial aesthetics and score significantly lower on orthognathic QoL measures compared to the general population [4,5]. When the severity of dentofacial disharmony exceeds what can be addressed through orthodontics alone, combined orthodontic-orthognathic surgical treatment is typically indicated. These interventions are extensive and demanding for both the patient and the healthcare system. To guide treatment eligibility, objective criteria such as the Index of Orthognathic Functional Treatment Need (IOFTN) have been developed [6].

The decision to undergo orthognathic surgery involves a complex interplay of medical, psychological, and social factors. These include emotional variables (e.g., fear, anxiety), financial considerations, scheduling flexibility, perceived social support, and interpersonal abilities such as communication and rapport building [7]. Dental professionals play a critical role in this process by raising awareness about the condition, explaining treatment options, coordinating interdisciplinary care, and managing patient expectations [8].

Studies have shown that patients' initial attitudes and willingness to undergo orthodontic treatment are strong predictors of compliance and pain tolerance often more so than demographic factors like age or socioeconomic status [9]. Key personality traits such as neuroticism, extraversion, conscientiousness, agreeableness, openness, perfectionism, and self-esteem are known to influence psychological functioning, daily decision-making, and attitudes toward surgical treatment [10]. However, the relationship between these traits and the motivation for orthognathic surgery remains underexplored.

This study aimed to explore how age and gender shape the relationship between personality traits, QoL, and a patient's decision to undergo surgery for correction of dentofacial deformities. The null hypothesis proposed that age and gender do not play a significant role on this relationship.

## Materials and methods

### Participants and study design

A cross-sectional study included 108 participants aged 14–53 years (median 18; interquartile range 17–26), of whom 68% were female. Participants were consecutively recruited from patients referred for orthodontic consultation and/or treatment of skeletal malocclusion at the University Dental Clinic Rijeka, Croatia, between September 1, 2020 and December 1, 2021.

Age groups were defined a priori: adolescents were ≤17 years (n = 46), and adults were ≥18 years (n = 62). This definition follows widely accepted clinical and legal cutoffs for adulthood.

Sample size determination was based on a power calculation using an online calculator https://www.danielsoper.com/statcalc. With α = 0.05, power = 0.80, and an expected hypothesized large effect size (Cohen's d = 0.8), the minimum sample required per group was n = 21 and 26 for one and two-tailed hypotheses. Groups were subjects who accepted vs. refused surgery, separately for two genders or two age groups, so in total 84–104 participants.

Inclusion criteria were: (1) moderate to very great need for surgery (grades 3–5) according to the IOFTN; (2) European origin (to ensure applicability of validated Croatian-language instruments); (3) absence of chronic systemic medical conditions; and (4) fluency in the Croatian language.

Exclusion criteria were: (1) age > 55 years; and (2) clinically significant neurodevelopmental or cognitive impairment that precluded valid questionnaire completion, as assessed by clinician judgment and medical history.

Consent procedures: For participants ≥18 years, written informed consent was obtained. For those <18 years, written informed consent from a parent or legal guardian, along with participant assent, was required.

### Adherence to the study protocol

The study was not preregistered. The study design, eligibility criteria, group definitions (adolescents vs. adults; acceptance vs. refusal of surgery), outcome measures, and statistical analyses were defined a priori and were adhered to throughout data collection and analysis.

Ethics approval: The study protocol was approved by the Ethics Committee of the University of Rijeka Faculty of Dental Medicine (Approval No. 2170-57-006-20-01; September 1, 2020) and the Clinical Hospital Center Rijeka (Approval No. 2170-29-02/1-20-2; July 12, 2020).

Data availability: Missing data did not exceed 5% per instrument. Cases with incomplete items were excluded pairwise; no imputation was applied. All anonymized individual level data underlying the results are now provided as Supporting Information files submitted with the revised manuscript. All anonymized data underlying the results are also available on the institutional repository (https://repository.fdmri.uniri.hr/).

### Instruments

**Orthognathic quality of life questionnaire (OQLQ).** The validated Croatian version of the OQLQ was used to capture four dimensions: *oral function (OF), facial esthetics concern (FE), social aspects of deformity (SA),* and

*awareness of dentofacial deformity (AW)* by using 22 items [11]. Each item was scored on a 4-point Likert scale, with higher scores indicating greater impairment. Internal consistency of Croatian OQLQ ranged from α = 0.80–0.92 [11].

**Big five inventory (BFI).** Personality traits were measured using the validated Croatian 44-item BFI [12], which assessed five broad personality dimensions with two facets for each trait: *extraversion (facets assertiveness and activity), agreeableness (altruism and compliance), conscientiousness (order and self-discipline), neuroticism (anxiety and depression),* and *openness to experience (esthetics and ideas)* [12,13]. Each item was rated on a 5-point Likert scale (1 = strongly disagree to 5 = strongly agree). The BFI has demonstrated robust psychometric properties across multiple cultural contexts, including European populations, with subscale internal consistency ranging between α = 0.70 and 0.85 [12].

**Frost multidimensional perfectionism scale (FMPS).** Perfectionism was assessed using the 35-item Croatian version of the FMPS [14]. The scale evaluated six dimensions: *personal standards, concern over mistakes, doubts about actions, parental expectations, parental criticism,* and *organization*. In addition, global perfectionism was calculated as a summary score of all dimensions except organization. Items were rated on a 5-point Likert scale (1 = strongly disagree to 5 = strongly agree). Previous studies in Croatian and European samples reported Cronbach's α values between 0.74 and 0.89 for subscales [14].

**Rosenberg self-esteem scale (RSES).** Global self-esteem was measured using the 10-item RSES, validated in Croatian population [15]. Each item was rated on a 4-point Likert scale (1 = strongly disagree to 4 = strongly agree), with higher scores reflecting greater self-esteem. The RSES has shown high internal reliability (α > 0.85) [15].

## Statistical analysis

All analyses were performed using IBM SPSS Statistics version 22 (IBM, Armonk, NY, USA). Data were first inspected for completeness, and missing responses within psychometric instruments were handled according to each scale's scoring manual. Participants with more than 10% missing items per instrument were excluded from the corresponding analysis. For cases with ≤10% missing data, mean substitution within the respective subscale was applied.

Data distribution was tested for normality using the Kolmogorov–Smirnov test and by visual inspection of histograms and Q–Q plots. Because most variables deviated from normality, non-parametric procedures were used. Descriptive statistics are reported as medians with interquartile ranges (IQR) for continuous variables and counts (percentages) for categorical variables.

The Mann–Whitney test was used to compare continuous variables between groups defined by surgical decision (acceptance vs. refusal), separately for adolescents and adults, and separately for males and females. Categorical variables were analyzed using Fisher's exact tests as appropriate with calculating odds ratio (OR). For all between-group comparisons, effect sizes were calculated to complement p-values. Formula $r = Z/\sqrt{N}$ was used for calculation of effect size in Mann-Whitney test, while Cramer's V was used for Fisher test. The following criteria was used for interpretation of r and V: < 0.3 low effect size, 0.3–0.5 moderate, 0.5–0.7 large and >0.7 very large. Hodges-Lehmann method was used to calculate median of differences with 95% confidence intervals (CI) in psychosocial dimensions between subjects who accepted and refused surgery.

Discriminant analysis was performed to assess the contribution of age, personality traits, and QoL to surgical decision-making across genders. Standardized canonical coefficients and structure matrix loadings were used to interpret variable contributions. The Wilks' lambda, canonical correlation, and classification accuracy (%) were reported. A two-tailed alpha level of 0.05 was set for statistical significance.

## Results

### Participant characteristics

Baseline characteristics are summarized in Table 1. The overall acceptance rate of orthognathic surgery was 51%.

**Table 1. Baseline demographic, clinical, and psychosocial characteristics of the study participants overall and according to age group and surgical decision. Continuous variables are presented as medians with interquartile ranges, and categorical variables as counts and percentages.**

| Characteristic | Total (n = 108) | Adolescents (≤17 years, n = 46) | Adults (≥18 years, n = 62) | Surgery accepted (n = 55) | Surgery refused (n = 53) |
|---|---|---|---|---|---|
| Age, years (median, interquartile range – IQR) | 18 (17 –26 ) | 17 (16 –17 ) | 23 (19 –30) | 20 (17 –27) | 17 (17 –22 ) |
| Female, n (%) | 73 (68%) | 35 (76%) | 38 (61%) | 35 (64%) | 38 (72%) |
| Male, n (%) | 35 (3%) | 11 (24%) | 24 (39%) | 20 (36%) | 15 (28%) |
| OQLQ – Oral function (median, IQR) | 6 (3 –10 ) | 5 (2 –9 ) | 8 (4 –12 ) | 9 (5 –13 ) | 4 (2 –8 ) |
| OQLQ – Facial aesthetics concern (median, IQR) | 12 (6 –16 ) | 13 (9 –15 ) | 12 (6 –16 ) | 14 (11 –16 ) | 11 (5 –14 ) |
| OQLQ – Social aspect (median, IQR) | 9 (3 –14 ) | 10 (6 –16 ) | 8 (1 –12 ) | 10 (7 –16 ) | 7 (2 –12 ) |
| OQLQ – Awareness of deformity (median, IQR) | 7 (4 –10 ) | 7 (5 –9 ) | 7 (3 –10 ) | 7 (4 –10 ) | 7 (3 –10 ) |
| Self-Esteem (median, IQR) | 43 (38 –46 ) | 41 (37 –45 ) | 43 (39 –46 ) | 41 (37 –45 ) | 43 (40 –47 ) |
| Global perfectionism (median, IQR) | 66 (54-81) | 66 (54-76) | 67 (54-83) | 72 (55-86) | 63 (54-74) |
| Extroversion (median, IQR) | 29 (23 –33 ) | 30 (24 –33 ) | 28 (23 –33 ) | 28 (23 –31 ) | 29 (24 –34 ) |
| Agreeableness (median, IQR) | 35 (32 –38 ) | 35 (33 –38 ) | 35 (31 –38 ) | 35 (31 –38 ) | 35 (33 –38 ) |
| Consciousnesses (median, IQR) | 34 (29 –37 ) | 33 (29 –36 ) | 34 (30 –37 ) | 34 (29 –37 ) | 34 (30 –37 ) |
| Neuroticism (median, IQR) | 20 (16 –23 ) | 19 (16 –24 ) | 20 (16 –22 ) | 20 (16 –24 ) | 19 (16 –22 ) |
| Openness (median, IQR) | 37 (33 –42 ) | 37 (32 –41 ) | 37 (33 –42) | 37 (31 –41 ) | 37 (33 –42 ) |

### Age effects on surgical decision-making

Adults were more likely to accept surgery than adolescents (61% vs. 37%; p = 0.019, V = 0.241; OR 2.7; 95% CI 1.2–5.9). Adults who accepted surgery had lower self-esteem than those that refused (median 43 vs. 45; p = 0.034, r = 0.270; median difference 2; 95% CI 0–5) and higher perfectionism (76 vs. 61; p = 0.012; r = 0.320; difference 13; 95% CI 3–22), particularly concern over mistakes (22 vs. 13; p < 0.001; r = 0.469; difference 7; 95% CI 3–11) and personal standards (24 vs. 20; p = 0.004; r = 0.370; difference 5; 95% CI 1–7). Among adolescents, personality traits showed no significant effect (Fig 1). However, in both age groups, impaired OF (8 vs. 3; p = 0.013; r = 0.368, difference 4; 95% CI 1–6 for adolescents and 10 vs. 6; p = 0.006; r = 0.348 for adults) and greater FE (14 vs. 11; p = 0.018; r = 0.350 for adolescents and 15 vs. 8; p = 0.015; r = 0.310; difference 4; 95% CI 1–7 for adults) influenced surgery acceptance.

### Gender effects on surgical decision-making

More males than females accepted surgery (57% vs. 48%,) but the difference was not statistically significant (p = 0.415). Males who accepted surgery had higher FE than those who refused (median 12 vs. 3; p = 0.009; r = 0.439; median difference 4; 95% CI 1–10), OF (9 vs. 2; p < 0.001; r = 0.597; difference 5; 95% CI 3–7), AW (7 vs. 2; p = 0.006; r = 0.463; difference 3; 95% CI 1–7) and concern over mistakes (21 vs. 12; p = 0.030; r = 0.367; difference 5; 95% CI 1–10, Fig 1). Females who accepted surgery had higher FE than those who refused (15 vs. 11; p = 0.006; r = 0.320; median difference 3; 95% CI 1–6), OF (10 vs. 6; p = 0.003; r = 0.342; difference 4; 95% CI 1–7), and concern over mistakes (21 vs. 16; p = 0.006; r = 0.325; difference 5; 95% CI 2–8), but lower self-esteem (40 vs. 44; p = 0.044; r = 0.235; difference 3; 95% CI 0–5).

### Simultaneous effects of gender, age, personality traits and QoL on surgical decision-making

Discriminant analysis identified two statistically significant canonical functions that accounted for 91.8% of the total variability in the data. The first function explained 55.7% of the variance (p < 0.001) while the second function explained 36.1% (p = 0.033; Table 2). Fig 2 illustrates the differences along the X-axis between female and male participants, which are less related to the decision to accept or refuse surgery. Women scored higher in organization, aesthetics, anxiety, and cooperation, and lower in ideas and parental expectations compared to men. On the Y-axis, the differences between individuals

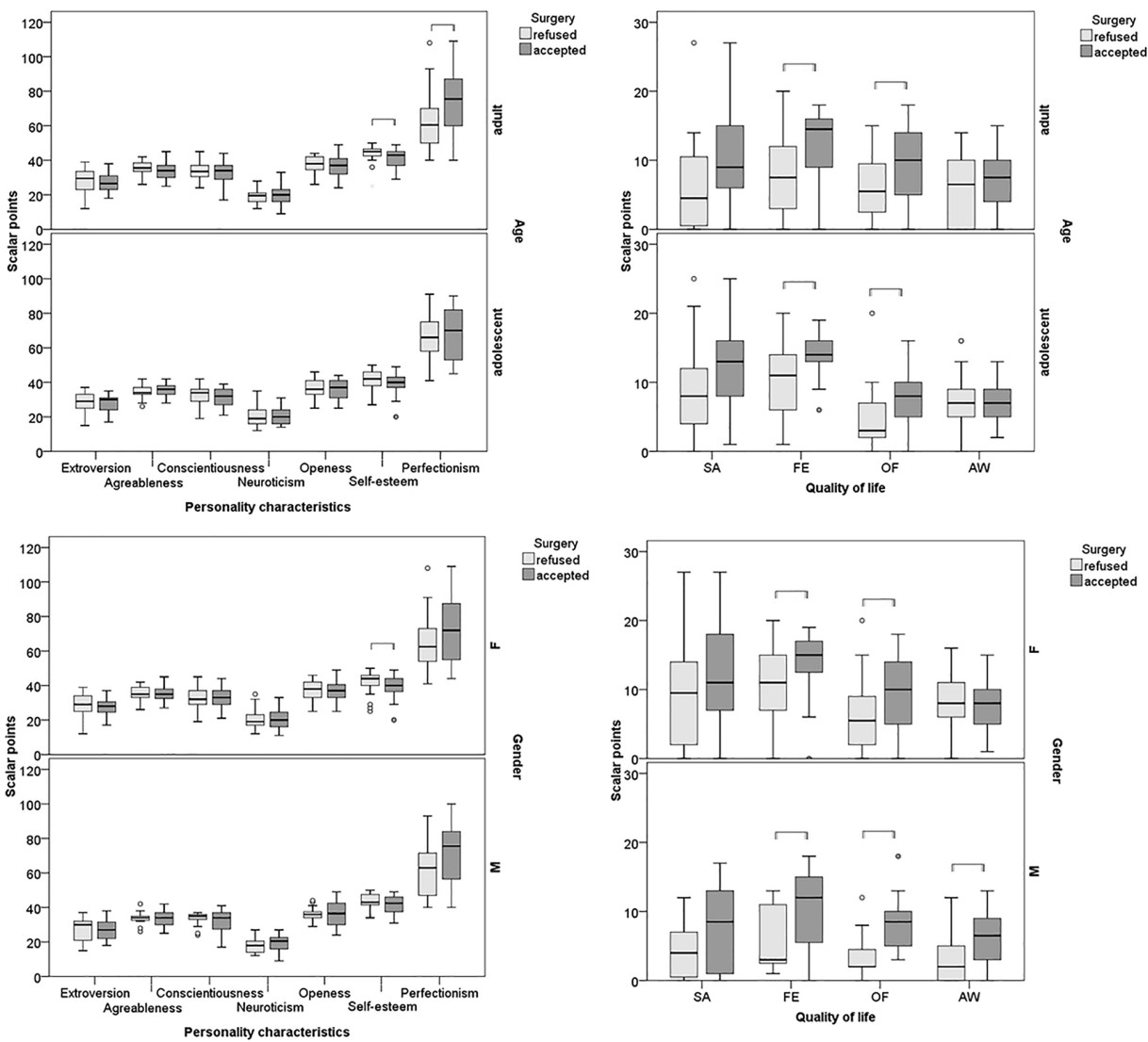

**Fig 1. Comparison of personality traits (left panels) and orthognathic quality of life dimensions (right panels) between participants who accepted and those who refused orthognathic surgery, stratified by age group (adolescents vs. adults) and gender (males vs. females).** Data are presented as medians with interquartile ranges. Statistically significant differences between acceptance and refusal groups within each stratum are indicated by parentheses (Mann–Whitney U test, $p < 0.05$).

who accepted and those who refused surgery were less dependent on gender. Participants who accepted surgery exhibited more impairment in OF and SA, higher FE, more concern over mistakes, greater personal standards, more doubts about their performance, higher parental complaints, more activity, higher order, and were older compared to those who refused surgery. Additionally, those who accepted surgery had lower self-esteem and assertiveness (Table 3).

## Discussion

The null hypothesis has been rejected. Overall, our findings indicate that age, as a marker of psychological development, has a more substantial influence on the relationship between personality traits, QoL, and decision to accept orthognathic surgery for dentofacial deformities than gender.

**Table 2. Eigenvalues, canonical correlations, and Wilks' lambda statistics for the discriminant functions derived from personality traits, quality of life dimensions, age, and gender in relation to surgical decision-making.**

| Function | Eigenvalue % | Variables% | Cumulatively% | Canonical correlation | Wilks lambda | p |
|---|---|---|---|---|---|---|
| 1 | 1.007 | 55.7 | 55.7 | 0.708 | 0.262 | <0.001 |
| 2 | 0.652 | 36.1 | 91.8 | 0.628 | 0.527 | 0.033 |
| 3 | 0.149 | 8.2 | 100.0 | 0.360 | 0.870 | 0.874 |

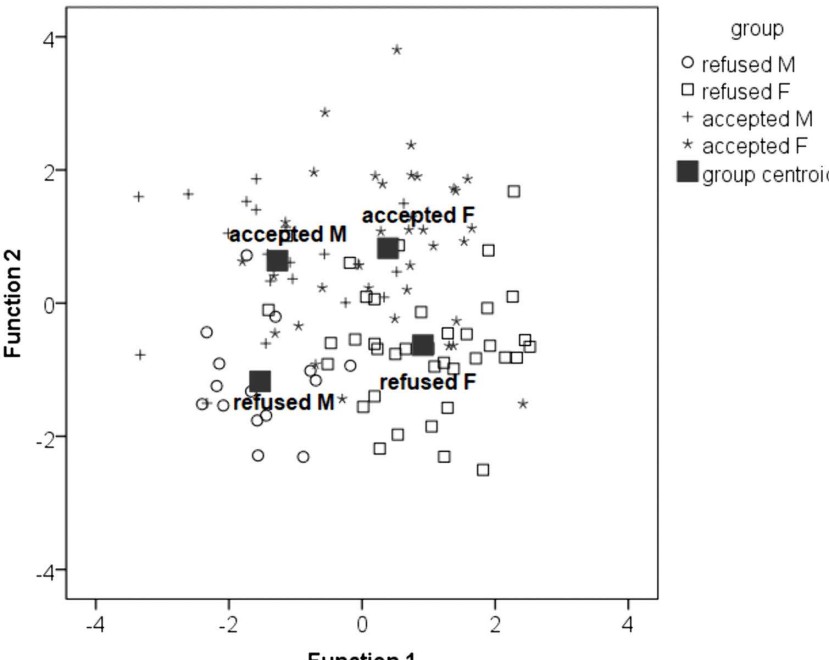

**Fig 2. Discriminant analysis plot showing the positions of group centroids for participants according to gender (males vs. females) and surgical decision (acceptance vs. refusal).** The axes represent the first two canonical discriminant functions. Distances between centroids reflect differences in the multivariate profiles of personality traits, quality of life dimensions, and age.

## Age and surgical decision

Adolescents were found to be less likely to accept combined orthodontic-surgical approach in managing their skeletal malocclusion compared to adults. During adolescence, facial features are still undergoing development, and the extent of deformities, such as mandibular prognathism may not yet be fully visible. Furthermore, adolescents may lack the psychological maturity required to manage the demands of surgery and recovery [16,17]. This developmental phase, which often coincides with important transitional life stages such as beginning university or entering the workforce, can contribute to the rejection of surgery due to the emotional and psychological strain these transitions already impose.

Research supports the notion that adults are more likely to seek aesthetic procedures, including orthognathic surgery, than adolescents [18,19]. By adulthood, individuals' facial characteristics are fully developed, making surgical outcomes more predictable and stable. Adults' motivations for undergoing surgery may stem from a desire to improve self-esteem or appearance, often driven by social or career pressures [20]. Additionally, life experience allows adults to better evaluate the long-term impact of dentofacial deformities, making them more informed in their decision-making. Adults typically

**Table 3. Structure matrix showing correlations between discriminant variables and canonical discriminant functions. Variables are ordered according to the absolute magnitude of their correlations within each function.**

|  | Function | | |
|---|---|---|---|
|  | 1 | 2 | 3 |
| Organization | 0.395* | 0.072 | −0.156 |
| Aesthetics | 0.329* | −0.141 | −0.084 |
| Anxiety | 0.210* | 0.155 | −0.066 |
| Ideas | −0.166* | 0.047 | 0.002 |
| Cooperation | 0.126* | −0.031 | 0.087 |
| Parental expectations | −0.077* | 0.052 | −0.054 |
| Oral function | 0.100 | 0.570* | −0.016 |
| Facial esthetic concern | 0.272 | 0.508* | −0.067 |
| Worry about mistakes | 0.014 | 0.466* | −0.119 |
| Social aspect of deformity | 0.242 | 0.344* | −0.178 |
| Personal standards | −0.081 | 0.268* | −0.022 |
| Self-esteem | −0.076 | −0.256* | 0.175 |
| Assertiveness | 0.039 | −0.201* | −0.026 |
| Doubts about their own performance | 0.027 | 0.198* | −0.014 |
| Activity | 0.105 | 0.195* | −0.117 |
| Age | −0.051 | 0.192* | 0.165 |
| Parental complaints | −0.048 | 0.089* | 0.048 |
| Order | 0.045 | 0.083* | −0.023 |
| Awareness of deformity | 0.393 | 0.207 | 0.415* |
| Altruism | 0.201 | 0.078 | −0.345* |
| Self-discipline | −0.006 | −0.070 | −0.300* |
| Depression | −0.008 | 0.082 | 0.291* |

* Highest absolute correlation between each variable and any discriminant function.

have greater financial independence, flexibility in scheduling, and the emotional maturity to navigate the challenges of surgery [21,22].

Our study found that personality traits, especially lower self-esteem and higher levels of perfectionism particularly in dimensions such as personal standards and concern over mistakes had a notable impact on adults' decisions to undergo surgery, but not adolescents'. Adults tend to have more stable self-esteem, while adolescents are in crucial stages of emotional and cognitive development and may be more susceptible to peer influence. Both perfectionism and self-esteem are shaped by genetic and environmental factors [14,15]. While adults are often motivated by a long-standing dissatisfaction with their appearance, adolescents may act impulsively or may not fully understand their underlying motives [23]. Research supports a link between neuroticism, perfectionism, and low self-esteem with the desire to undergo aesthetic procedures, although traits such as introversion and extraversion tend to show weaker associations [24].

The decision to undergo orthognathic surgery was influenced by concerns about both OF and FE in both adolescents and adults. Impaired ability to bite, chew, and function without pain is highly valued by both groups, as functional impairments directly affect overall well-being. Some evidence even suggests that functional concerns may be stronger motivators than aesthetic or social factors [25,26]. Furthermore, societal beauty standards, social media, and exposure to idealized images can influence both age groups. For adolescents, social acceptance may hinge on physical attractiveness, while for adults, appearance can impact professional and social progression [9,27–31].

Some studies, however, suggest that younger individuals may be more motivated by body image concern than older individuals, possibly due to greater societal pressure on adolescents to conform to idealized beauty standards promoted by peers and the media [32,33]. In contrast, our findings suggest that functional considerations may play a more prominent role in surgical decision-making, particularly across age groups.

## Gender and surgical decision

In both males and females, higher FE, OF, and concern over mistakes (perfectionism dimension) were motives to accept surgery. The only surgical motives that were different in genders were higher AW in males and lower self-esteem in females. Discriminant analysis revealed that women had higher levels of organization, anxiety, and aesthetic sensitivity, while they reported lower parental expectations compared to men. These findings may support the idea that societal pressures on women to conform to aesthetic ideals remain pronounced, while they are often held to different standards of performance in other areas compared to men [34–37]. Although the prevalence of dentofacial deformities is similar across genders, women tend to seek consultations and utilize healthcare services more often than men [25]. This could suggest that FA may be more prominent motivator among women, though this was not corroborated by our findings.

Discriminant analysis also showed that, irrespective of gender, individuals who accepted surgery were more influenced by impaired OF, social difficulties, FE, perfectionism, and lower self-esteem. Factors such as age, conscientiousness, and extraversion had a lesser impact. These results align with previous research that highlights the importance of functional and psychosocial factors, including family pressure and social expectations, over demographic factors in surgical decision-making [35,36]. Sociocultural factors may also shape how patients perceive and respond to such procedures. Adults tend to pursue surgery more frequently, despite the higher costs associated with treatment, suggesting that financial concerns are less of a barrier for them [37–39].

## Strengths and limitations

One of the strengths of this study is its use of a multivariate model to control for multiple variables, providing a more nuanced understanding of the factors influencing surgical decisions. However, several limitations should be acknowledged. First, the sample size was modest, particularly for male participants, limiting the generalizability of gender-related findings. Although the a priori power calculation supported adequacy for detecting large effects, larger samples are required to reliably detect small-to-moderate effects, to enable adequately powered subgroup analyses (particularly among males), and to improve the stability of multivariable models including multiple psychosocial predictors. Second, all participants were of European origin, reflecting the validated language versions of our instruments; this reduces applicability to more diverse populations. Third, the study was not preregistered, which we recognize as a methodological limitation. Nevertheless, the study design, group definitions, outcome measures, and planned statistical analyses were defined a priori and were adhered to throughout data collection and analysis, with no post hoc modifications to the primary study setup. Fourth, while validated instruments were used, other relevant psychosocial variables, such as body image and socioeconomic status, were not assessed. Finally, the cross-sectional design precludes causal inference.

## Implications and future directions

Despite these limitations, the study highlights that developmental stage plays a greater role than gender in shaping the relationship between psychosocial aspects and surgical acceptance. These insights are clinically relevant for orthodontists and maxillofacial surgeons, who should tailor communication and counseling according to patients' age, psychological profile, and functional concerns. Future research should include larger and more diverse samples, preregistered study designs, and a broader range of psychological and socioeconomic variables to better capture the multifaceted nature of surgical decision-making.

## Conclusion

The relationship between personality traits, QoL, and the decision to undergo orthognathic surgery was more strongly influenced by age than by gender. While OF and FE influenced acceptance across all age and gender groups, personality-related factors were particularly relevant among adults.

## Supporting information

**S1 File. Supporting Information files.**
(ZIP)

## Author contributions

**Conceptualization:** Renata Vidakovic, Stjepan Spalj.

**Data curation:** Renata Vidakovic, Martina Zigante.

**Formal analysis:** Renata Vidakovic.

**Funding acquisition:** Renata Vidakovic, Stjepan Spalj.

**Investigation:** Martina Zigante.

**Methodology:** Renata Vidakovic, Stjepan Spalj.

**Resources:** Renata Vidakovic.

**Software:** Renata Vidakovic.

**Supervision:** Stjepan Spalj.

**Validation:** Renata Vidakovic.

**Visualization:** Jeta Kelmendi, Martina Zigante.

**Writing – original draft:** Jeta Kelmendi.

**Writing – review & editing:** Stjepan Spalj.

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
