## [Decision Letter · Decision Letter 0]

15 Sep 2025

PONE-D-25-27428IS THE DECISION ABOUT SURGICAL TREATMENT OF DENTOFACIAL DEFORMITY RELATED TO GENDER AND UNDER DIFFERENT INFLUENCES IN ADOLESCENTS AND ADULTS?PLOS ONE?

Dear Dr. Kelmendi,

We look forward to receiving your revised manuscript.

Kind regards,

Andrej M Kielbassa

Academic Editor

PLOS ONE

Journal Requirements:

- https://doi.org/10.1093/ejo/cjac060

In your revision ensure you cite all your sources (including your own works), and quote or rephrase any duplicated text outside the methods section. Further consideration is dependent on these concerns being addressed.

“none”

4. We note that your Data Availability Statement is currently as follows: All relevant data are contained within the manuscript and its Supporting Information files.

5. Please note that your Data Availability Statement is currently missing [the repository name and/or the DOI/accession number of each dataset OR a direct link to access each database]. If your manuscript is accepted for publication, you will be asked to provide these details on a very short timeline. We therefore suggest that you provide this information now, though we will not hold up the peer review process if you are unable.

Reviewers' comments:

Reviewer's Responses to Questions

**Comments to the Author**

1. Is the manuscript technically sound, and do the data support the conclusions?

Reviewer #1: Partly

Reviewer #2: Yes

Reviewer #3: Yes

2. Has the statistical analysis been performed appropriately and rigorously?

Reviewer #1: No

Reviewer #2: Yes

Reviewer #3: Yes

3. Have the authors made all data underlying the findings in their manuscript fully available?

Reviewer #1: No

Reviewer #2: Yes

Reviewer #3: Yes

4. Is the manuscript presented in an intelligible fashion and written in standard English?

Reviewer #1: Yes

Reviewer #2: Yes

Reviewer #3: Yes

Reviewer #1: General notes

- The study presents the results of original research. - Yes

- Results reported have not been published elsewhere. - Yes

- Experiments, statistics, and other analyses are performed to a high technical standard and are described in sufficient detail. - No, there minor issues

- Conclusions are presented in an appropriate fashion and are supported by the data. - No

- The article is presented in an intelligible fashion and is written in standard English. - Yes

- The research meets all applicable standards for the ethics of experimentation and research integrity. - No

- The article adheres to appropriate reporting guidelines and community standards for data availability. - No

Title

- "IS THE DECISION ABOUT SURGICAL TREATMENT OF DENTOFACIAL DEFORMITY RELATED TO GENDER AND UNDER DIFFERENT INFLUENCES IN ADOLESCENTS AND ADULTS?" No capital letters, please. Stick to the Guidelines for Authors, and consult some recently published Plos One papers.

- No questions as Title. Revise thoroughly.

- Type of study must be provided with your Title.

Abstract

- Please note that Plos One allows for a maximum word count of 300 words. With your current 195 words, you do not provide maximum information. Please revise carefully, but thoroughly.

- "This study investigated how age and gender influence the relationship between personality traits, quality of life (QoL), and a patient’s decision to accept orthognathic surgery." Remember to adapt aims to your full text.

- Provide exact results, and give exact P values.

- With your Conclusions, please stick exclusively to your revised aims. Do not simply repeat your results here. Do not speculate. Do not present meaningless phrases. Instead, provide a reasonable and generalizable extension of your outcome.

- This section would not seem convincingly elaborated, and re-review is considered mandatory. Please remember that this section in particular is important, since readers will switch to your full text AFTER having checked your Abstract.

Intro

- With your headlines, please stick to Journal style. "INTRODUCTION" must read "Introduction". Consulting some recently published Plos One papers would seem helpful.

- You have studied several different aspects. Please provide sound and valid null hypotheses. Note that your current "hypothesis" is not acceptable.

Meths

- "This cross-sectional study included a total of 108 participants, (...)." Why did you include 108 participants?

- What about a sound sample size calculation? Sentences like "A priori sample size estimation was performed and previously reported by Vidakovic et al. [11]." would not seem acceptable, no doubt. Please see comments on H0 as given above, provide a sound calculation, and explain.

- Cross-sectional studies are observational studies. What about the pre-registration of your study? Please go to https://pmc.ncbi.nlm.nih.gov/articles/PMC2952011/, and explain your rationale.

- "Caucasian ethnicity" would not be acceptable. There's no scientific justification for use of that term (referring actually to a 19th-century anthropological idea that was based around a false conception).

- What does "culturally adapted following COSMIN guidelines" mean? Please provide deatils.

- "Big Five Inventory" would call for a sound reference. Please provide explanatory details.

- Same with of "Frost Multidimensional Perfectionism Scale (FMPS)".

- Same with "Rosenberg Self-Esteem Scale (RSES)".

- Do not use legal terms like "inc.", and so on.

Results

- Why do you stick to "(p ≤ 0.034)"? "≤" would not seem clear, please revise carefully.

- Please revise you Figures, to ensure reproducibility.

- Remember to provide as much details as possible with your Figures and Tables. The latter must be sef-explanatory.

Disc

- What about H0? Was your null hypothesis rejected or not rejected?

- What about the low number of participants?

- What about the generalizability of your outcome?

- What about the responses to your teaching objectives?

Concl

- Again, with your Conclusions, please stick exclusively to your revised aims. Do not simply repeat your results here. Do not speculate (such thoughts will be better given with your Disc section). Instead, provide a reasonable and generalizable extension of your outcome.

In total, this submitted draft might be interesting, but several questions as given above must be answered first. Several typos (for example, "FOUNDING") must be revised, and re-review of a re-submitted draft will be mandatory.

Reviewer #2: Congratulations to the authors on their work. The writing is clear, and the study fosters a meaningful discussion on an important aspect of dentofacial deformity treatment. The research is interesting and addresses a clinically significant question, with strengths including the use of validated instruments and a multivariate approach. The sample size is predominantly female, which can be considered a limitation. The conclusions regarding the "minimal influence" of gender may be partly due to the small number of male participants, making it difficult to detect a real effect if one exists. Nevertheless, this likely does not substantially compromise the overall results.

Reviewer #3: • Resolve funding / disclosure inconsistency. The submission metadata and manuscript text present conflicting statements about funding (one place says “no specific funding” while the manuscript later lists a University of Rijeka grant). Authors must provide a single, accurate funding statement (grant number, funder name) and explicitly state the funder’s role (or lack thereof) in study design, analysis, and manuscript preparation.

• Define age groups explicitly. State the exact age cutoff used to define “adolescents” and “adults” (e.g., ≤17 or ≤18 for adolescents). Report the n in each age group and show those numbers in tables/figures.

• Describe consent for minors. For participants <18, explicitly state that parental/guardian informed consent and participant assent were obtained and how this was managed.

• Clarify and harmonize inclusion/exclusion criteria language. Replace the vague term “mental disabilities” with a precise definition (e.g., “clinically significant neurodevelopmental or cognitive impairment that precludes valid completion of questionnaires”), and state how this was assessed (medical record, clinician screening, standardized instrument).

• Reconcile ethnicity restriction and explain rationale. Authors restricted enrollment to “Caucasian ethnicity.” Provide a clear rationale for this choice, describe how ethnicity was determined, and discuss implications for generalizability in the Limitations.

• State how missing data were handled. Provide counts of missing items per instrument and describe any imputation or exclusion rules applied.

• Report effect sizes and 95% confidence intervals. For all main comparisons (adults vs adolescents; accepters vs refusers; key discriminant predictors) include effect sizes (Cohen’s d, eta², or equivalent) and 95% CIs, not p-values alone.

• Provide a complete baseline characteristics table. Include age (median/IQR or mean/SD), sex, IOFTN grades distribution, socioeconomic indicators (if available), and main scores (OQLQ subscales, BFI domains, FMPS, RSES) stratified by surgery acceptance and by age group. Make sample sizes clear.

• Clarify the role of gender. The abstract and text state “gender showed minimal effect” and “not influenced by gender.” Reword to “no statistically significant association with gender was observed,” and show the test results (p, effect size) that support this.

**Do you want your identity to be public for this peer review?** For information about this choice, including consent withdrawal, please see our Privacy Policy

Reviewer #1: No

Reviewer #2: No

Reviewer #3: No

---

## [Author Response · Author response to Decision Letter 1]

19 Nov 2025

Dear Editor,

Thank you very much for the opportunity to revise our manuscript. We carefully considered all comments from the reviewers and the editor. Each point has been addressed in detail in the attached Point-by-Point Response Letter.

The revised version includes updated statistical analyses (including Wilks’ lambda, canonical correlation, effect sizes, and 95% confidence intervals), clarification of the methods, and reorganization of the tables and figures according to PLOS formatting requirements.

We greatly appreciate the constructive feedback, which helped us to substantially improve the accuracy, transparency, and clarity of our manuscript.

Kind regards,

Jeta Kelmendi

Corresponding Author

---

## [Decision Letter · Decision Letter 1]

28 Nov 2025

Dear Dr. Kelmendi,

Thank you for submitting your manuscript to PLOS ONE. After careful consideration, we feel that it has merit but does not fully meet PLOS ONE’s publication criteria as it currently stands. Therefore, we invite you to submit a revised version of the manuscript that addresses the points raised during the review process.

We look forward to receiving your revised manuscript.

Kind regards,

Andrej M Kielbassa

Academic Editor

PLOS ONE

Journal Requirements:

Reviewers' comments:

Reviewer's Responses to Questions

**Comments to the Author**

Reviewer #1: All comments have been addressed

Reviewer #2: All comments have been addressed

2. Is the manuscript technically sound, and do the data support the conclusions?

Reviewer #1: No

Reviewer #2: Yes

3. Has the statistical analysis been performed appropriately and rigorously?

Reviewer #1: No

Reviewer #2: Yes

4. Have the authors made all data underlying the findings in their manuscript fully available?

Reviewer #1: No

Reviewer #2: Yes

5. Is the manuscript presented in an intelligible fashion and written in standard English?

Reviewer #1: Yes

Reviewer #2: Yes

Reviewer #1: With the help of the Reviewers, this revised and re-submitted draft has been considerably improved. Unfortunately, still some further revisions would seem mandatory, please see comments given below.

Title

- Again, no capital letters, please. See previous comments.

(- Same with your subheadings – no capital letters, please. Revise thoroughly.)

Abstract

- What does “median 18” mean? Miles? Light years? Years? Please revise carefully.

Meths

- “median 18; interquartile range 17–26” – see comments given above.

- Again, provide dates of your ethical approvals.

- “With α = 0.05, power = 0.80, and an expected large effect size (Cohen’s d = 0.8), the minimum sample required per group was (...).” Rationale still would seem unclear. Why did you expect a large effect size? Please revise carefully, and provide your a-priori thoughts.

- Again, information on pre-registration is missing. You have indiacted that “(…) the study was not preregistered, which we recognize as a methodological limitation”. Consequently, please add information here.

- Again, please consult some recently published Plos One papers. You will not find any in-between bold formatting. Revise carefully, and stick to sound headlines. No in-between bold characters.

- “All anonymized data underlying the results are available on institutional repository (https://repository.fdmri.uniri.hr/).” Please stick to international standards, and provide data as additional files. You might wish to call this file “Appendix”. Remember that "The PLOS Data policy requires authors to make all data underlying the findings described in their manuscript fully available without restriction".

- Please delete “Corp.”.

Results

- “More males than females accepted surgery (57% vs. 48%) but the difference was not statistically

significant.” Again , please provide exact P value.

- With both Figures and Tables, please provide sound and satisfying legend. Remember that each Fig/Table must be self-explanatory, without any needs to stick to your full text.

Disc

- Please compare “OF and FE primarily influence decision, regardless of age or gender.” (Abstract section) and “This study reveals that age (…) has a more substantial influence (…) than gender.” (Disc section). Please adapt carefully, and revise.

- “Third, the study was not preregistered, which we recognize as a methodological limitation.“ Please provide a valid statement on whether you have followed your primary study set-up.

- Same with “Future research should include larger and more diverse samples, (...)”. Please explain why YOU have not included a broader sample size. This would have been an easy task, you surely will agree.

- Same with your sample size calculation. In case you are sure that “n=21 and 26 for one- and two-tailed hypotheses” would be sufficient, why do you think that “larger and more diverse samples” would seem mandatory? Remember that your "manuscript must describe a technically sound piece of scientific research with data that supports the conclusions. Experiments must have been conducted rigorously, with appropriate controls, replication, and sample sizes".

Concl

- Again, stick exclusively to your aims here.

Refs

- Please stick to Journal style, and provide exact information. See, for example, “Available from: UCL Discovery”. Information as given is not considered sufficient. Rememeber that neither the in-house editor nor the typesetter will be able to add such information to your draft. Providing sound mansucripts is considered the Co-Authors’ task, and only sound mansucripts will be acceptable.

Reviewer #2: (No Response)

**Do you want your identity to be public for this peer review?** For information about this choice, including consent withdrawal, please see our Privacy Policy

Reviewer #1: No

Reviewer #2: No

---

## [Author Response · Author response to Decision Letter 2]

24 Dec 2025

All reviewer and editor comments have been addressed in full. A detailed point-by-point response is provided in the uploaded “Response to Reviewers” document, dataset is uploaded as Supporting Information and all changes are highlighted in the revised manuscript with track changes.

---

## [Decision Letter · Decision Letter 2]

28 Dec 2025

The role of age and gender in the relationship between personality traits, quality of life, and decision-making about orthognathic surgery – a cross-sectional study

PONE-D-25-27428R2

Dear Dr. Kelmendi,

We’re pleased to inform you that your manuscript has been judged scientifically suitable for publication and will be formally accepted for publication once it meets all outstanding technical requirements.

Kind regards,

Prof. Dr. Dr. h. c. Andrej M Kielbassa

Academic Editor

PLOS One

Additional Editor Comments (optional):

Reviewers' comments:

Reviewer's Responses to Questions

**Comments to the Author**

Reviewer #1: All comments have been addressed

2. Is the manuscript technically sound, and do the data support the conclusions?

Reviewer #1: Yes

3. Has the statistical analysis been performed appropriately and rigorously?

Reviewer #1: Yes

4. Have the authors made all data underlying the findings in their manuscript fully available?

Reviewer #1: Yes

5. Is the manuscript presented in an intelligible fashion and written in standard English?

Reviewer #1: Yes

Reviewer #1: The Co-Authors have addressed all previous comments and recommendations. With the help of the external reviewers, this revised and re-submitted has been considerably improved, and would seem ready to proceed.

**Do you want your identity to be public for this peer review?** For information about this choice, including consent withdrawal, please see our Privacy Policy

Reviewer #1: No

---

## [Editor Report · Acceptance letter]

PONE-D-25-27428R2

PLOS One

Dear Dr. Kelmendi,

I'm pleased to inform you that your manuscript has been deemed suitable for publication in PLOS One. Congratulations! Your manuscript is now being handed over to our production team.

Kind regards,

on behalf of

Prof. Dr. med. dent. Dr. h. c. Andrej M Kielbassa

Academic Editor

PLOS One